# *GrAb*: A Deep Learning-Based Data-Driven Analytics Scheme for Energy Theft Detection

**DOI:** 10.3390/s22114048

**Published:** 2022-05-26

**Authors:** Sudeep Tanwar, Aparna Kumari, Darshan Vekaria, Maria Simona Raboaca, Fayez Alqahtani, Amr Tolba, Bogdan-Constantin Neagu, Ravi Sharma

**Affiliations:** 1Department of Computer Science and Engineering, Institute of Technology, Nirma University, Ahmedabad 382481, India; 16bit042@nirmauni.ac.in; 2Institute of Computer Technology, Ganpat University, Ahmedabad 384012, India; abk03@ganpatuniversity.ac.in; 3National Research and Development Institute for Cryogenic and Isotopic Technologies—ICSI Rm. Valcea, Uz-Inei Street, No. 4, Raureni, P.O. Box 7, 240050 Rm. Valcea, Romania; 4Software Engineering Department, College of Computer and Information Sciences, King Saud University, Riyadh 12372, Saudi Arabia; fhalqahtani@ksu.edu.sa; 5Computer Science Department, Community College, King Saud University, Riyadh 11437, Saudi Arabia; atolba@ksu.edu.sa; 6Department of Power Engineering, “Gheorghe Asachi” Technical University of Iasi, 700050 Iasi, Romania; bogdan.neagu@tuiasi.ro; 7Centre for Inter-Disciplinary Research and Innovation, University of Petroleum and Energy Studies, Dehradun 248007, India; ravisharmacidri@gmail.com

**Keywords:** deep learning, demand response management, energy consumption prediction, energy theft, LSTM, smart grid

## Abstract

Integrating information and communication technology (ICT) and energy grid infrastructures introduces smart grids (SG) to simplify energy generation, transmission, and distribution. The ICT is embedded in selected parts of the grid network, which partially deploys SG and raises various issues such as energy losses, either technical or non-technical (i.e., energy theft). Therefore, energy theft detection plays a crucial role in reducing the energy generation burden on the SG and meeting the consumer demand for energy. Motivated by these facts, in this paper, we propose a deep learning (DL)-based energy theft detection scheme, referred to as *GrAb*, which uses a data-driven analytics approach. *GrAb* uses a DL-based long short-term memory (LSTM) model to predict the energy consumption using smart meter data. Then, a threshold calculator is used to calculate the energy consumption. Both the predicted energy consumption and the threshold value are passed to the support vector machine (SVM)-based classifier to categorize the energy losses into technical, non-technical (energy theft), and normal consumption. The proposed data-driven theft detection scheme identifies various forms of energy theft (e.g., smart meter data manipulation or clandestine connections). Experimental results show that the proposed scheme (*GrAb*) identifies energy theft more accurately compared to the state-of-the-art approaches.

## 1. Introduction

Over the years, smart electrical appliances have become an essential part of our day-to-day activities that offer consumers a fully easy-to-use automated environment and utility savings. In addition, transforming the traditional grids into smart grids (SG) requires efficient analysis and management of energy consumption to balance the demand and supply of energy [1]. Moreover, it benefits the society in terms of efficient energy usage and conservation [2,3]. According to the Electric Power Research Institute (EPRI), the energy department has predicted only a 1% annual growth rate in electricity consumption for the year 2008–2035 [4]. Although the SG installation is considerably expensive, its benefits have proved it to be highly worthy, as more and more countries are keen on expanding their use of SG in various sectors such as smart cities development, smart transportation, and many more [5]. The expansion of the SGs can be correlated to its predicted market size from the year 2017 to 2023, as shown in Figure 1 [6].

Despite several essential capabilities of SG, it comes with several challenges, such as technical and non-technical losses of energy, data delivery, support for reliability, and security [7]. For example, technical energy loss is identified as facing an outage, faulty meters, or inefficient hardware implementation. On the other hand, the non-technical losses, commonly known as energy theft, are detected when the forger uses unethical practices, for instance, manipulating the energy load consumption data by modifying the energy reading of the smart meter. As a result, energy theft has become a severe concern in some countries, proliferating at an alarming rate that directly affects the nation’s economy. For instance, Northeast Group, a Limited Liability Company (LLC), mentioned that the world loses US$89.3 billion annually to electricity theft in one of the annual study reports. India tops the list with an annual loss of $16.2 billion, followed by Brazil and Russia [8,9].

To handle the aforementioned issues of energy theft, various research work has been done by the scientific community across the globe [10]. The emergent technology of deep learning (DL) has become the forerunner when it comes to energy management and pattern analysis of demand–supply of energy and theft detection in this economy [11,12,13,14]. A recent study conducted by Harvard University argued that artificial intelligence (AI) will revolutionize the energy industry and growth in energy production can be foreseen if we employ DL algorithms for energy management in the near future as shown in Figure 2 [1,15,16,17].

However, energy theft is becoming a major concern in the energy generation and management within all societies around the world [18,19]. On a residential basis, people are adopting fraudulent activities to cut down the load consumption of electrical appliances, resulting in energy theft prospects in SG [20,21]. There are various malpractices on the demand-side to tamper with the energy consumption readings, such as installing their own pipes to bypass the smart meter so that the energy supply does not get measured, damaging, or removing the teeth of the cog used to measure energy and tamper the readings manually. On large-scale operations, it has been seen that streets have been dug up to access the main supply. By doing so, people not only commit unethical practices and crimes but they also leave themselves and those around prone to burns, shocks, fire, and explosions. Therefore, it becomes indispensable to detect outlier changes in energy consumption reading to commercial buildings, residential houses, or industries to prevent any technical or non-technical fault in the future [22,23,24,25].

In this work, we propose a DL-based long-short term memory (LSTM) model for energy consumption prediction, which is further used to compare with the actual threshold consumption values obtained from smart meters. Then, the difference between predicted energy consumption and threshold values decides if there has been tampered data in the smart meter reading [26]. If the actual energy consumption and the predictions are within the threshold value range, it is considered a normal consumption scenario; otherwise, it is an act of energy theft. Additionally, Table 1 comprises list of the symbols and its abbreviation used during implementation of the *GrAb* model.

### 1.1. Research Contributions

The following are research contributions provided by the proposed *GrAb* model:Proposed an LSTM load prediction model to support accurate predictions of time series data provided by the smart meters installed in the smart home.Provided an energy loss classification mechanism consisting of an LSTM-SVC-based deep learning approach, which can help in detecting a loss of energy taking place in the smart grid system. In case the energy loss is there, then it is classified as technical loss or non-technical loss, i.e., energy theft.Comparative analysis of the LSTM model and the loss classification technique with other state-of-the-art approaches to evaluate the performance of the *GrAb*.

### 1.2. Organization of the Paper

This paper highlights the importance of smart grids, energy load prediction, energy losses, energy theft detection, demand response, and DL methods, especially LSTM. In Section 2, the data-driven schemes are described to detect the energy losses that can be further used to perceive energy theft. Then, the system model and problem formulation of *GrAb* are elaborated in Section 3. Next, Section 4 discusses the workflow of the proposed *GrAb* model. Then, Section 5 deals with the experimentation and results obtained by the *GrAb* model. Finally, Section 6 makes a conclusion drawn after carrying out the studies and experiments of this paper.

## 2. Related Work

Energy theft usually occurs when someone unethically tampers the readings of his smart meters to show a lesser energy usage than usual. As a result, the tampered readings are recorded and considered as consumption. Therefore, it becomes essential to detect such energy theft so that no one can take unethical advantage. It can be accomplished by monitoring consumer behavior toward their energy usage; if there is a sudden deviation in energy usage, it can be considered an outlier (forger). Here, in case the deviation is directed toward less energy usage, it may raise the probability of energy theft [27,28].

Earlier, there have been a few attempts to explore the issue of energy theft detection and loss classification for SG systems. In the work proposed by Yip et al. [29], there has been a focus on reducing non-technical loss due to energy thefts and defects in smart meters. In this study, the authors have proposed two linear regression-based algorithms called linear regression-based schemes for detecting energy theft and defective smart meters (LR-ETDM) and categorical variable-enhanced linear regression-based scheme for detecting energy theft and defective smart meters (CVLR-ETDM). The LR-ETDM has been presented based on consistent energy theft. Therefore, to deal with inconsistent patterns in energy theft, they deployed categorical variables into linear regression and developed CVLR-ETDM to detect even inconsistent theft patterns and faulty equipment in the SG system.

Furthermore, DL approaches have been used to improve the accuracy of the predictions. Li et al. [30] proposed a convolutional neural network-random forest (CNN-RF) model for detection of electricity theft. They have designed a CNN model to learn the various features of the dataset for different hours and days by using convolution and downsampling. They have used a dropout of 0.4 to avoid overfitting. Over the obtained features, the random forest model (RFM) is used to identify if there is energy theft or not. They carried out the experimentation on the sustainable energy authority of Ireland (SEAI) and Low-Carbon London (LCL) datasets. However, the model does not support load forecasting, and it only considers the historical data of smart meters.

Various methods have been deployed using the CNN mechanisms; Hasan et al. [31] has used the CNN-LSTM based DL approach for energy theft detection. They deployed the synthetic minority over-sampling technique (SMOTE) to generate new data points in the dataset to add the theft users and handle the class imbalance problem between normal users and theft users. The technique obtained 89% classification accuracy over the data provided by the State Grid Corporation of China. They used the confusion matrix as a parameter to rate their model. The shortcoming with this approach is that the proposed mechanism used a dataset that has no time attribute. Hence, this approach can not catch the trends in the dataset according to the time of use. The whole essence of theft detection is based on the time of a day/month and corresponding load consumption at that specific time. Then, Jindal et al. [32] presented an approach using k-means clustering to analyze the normal load consumption and the anomalous load consumption over the Dutch Residential Energy Dataset. This approach detects theft based on the difference between the predicted load and the meter reading data. However, the threshold calculation of (i.e., 10%) to energy consumption to detect energy theft raises the concern for its applicability in real life.

Yan et al. in [33] have proposed an AI-based ensemble method to detect the abnormal activities in the electricity metering infrastructure. They have used the Irish smart energy trails dataset, which has both normal and malicious data on which their ensemble classifiers are trained to bifurcate them. Their proposed approach outperforms other classifiers such as support vector machines, decision trees, and many more in terms of accuracy and false-positive rate. Then, the authors of [34] presented a deep neural network-based electricity theft detection in SGs. They discuss feature space importance, weaknesses in data pre-processing steps, and class imbalance problems. To overcome the above-mentioned issues, they compare and analyze the dataset with dynamic feature space by employing principal component analysis. Furthermore, to optimize the classifier accuracy, they used hyperparameters of the Bayesian optimizer. Their results outperform other baseline works in terms of accuracy and area under the curve (AUC) score. However, their contribution is restrictive to the dataset; hence, it is not adaptable in real-time scenarios. Furthermore, Wen et al. in [35] discuss various challenges of energy theft detection in SG, such as centralized system, privacy, and security. To tackle the aforementioned challenges, they have used distributed systems such as federated learning to preserve the privacy of the energy consumption data. To train the federated model, they have used homomorphic encryption-based encrypted training parameters instead of the raw data of SG. Then, to effectively detect the energy theft, they employed a temporal convolutional network with higher accuracy than the other state-of-the-art solutions. Table 2 shows the comparative analysis between the existing state-of-the-art work with the proposed work.

The authors of [36] studied the security aspect of energy consumption, where SGs incur from the unethical practices of electricity theft. To resolve the security issues, the authors have adopted the DL-based stacked autoencoders and sampling methods, where autoencoders are trained to extract feature space of energy consumption; then, sampling methods have been applied to solve the imbalance problem of the dataset. The results show that their proposed scheme outperforms the existing baseline work in terms of accuracy and detection rate. However, the aforementioned literature is not entirely exploring the potential extent of energy theft. Then, Godahewa et al. in [37] introduced an intelligent framework that predicts the temperature of the room and improves their setpoints. They have used a time series dataset to train the neural network that effectively predicts the temperature and sets the baseline when the room is occupied or unoccupied. Their proposed trained models save energy up to 15–20% approximately compared to the conventional temperature control system. However, the authors have not investigated the security prospect of their proposed framework. Then, the authors of [38] proposed a novel prediction technique for household power consumption prediction. They have employed AI-based models along with discrete transform function, i.e., wavelet transform. Their technique involves a self-attenuation mechanism that quickly learns the composite patterns of the household power consumption data. However, the proposed mechanism is not evaluated against any security attack such as data integrity attacks (load consumption data is modified by the attacker) that jeopardize the performance of the proposed mechanism. Furthermore, Zhenzhi et al. in [39] study different technologies to detect the electricity theft at the consumer side. They discuss existing baseline works, advantages and disadvantages of electricity theft detection technologies, and challenges in theft detection.

Several research works exist in the literature; still, the energy theft detection has not been explored to its full potential [40,41,42,43]. Hence, we propose the *GrAb* model to address all the challenges and important features for consideration to carrying out energy theft detection. Any variation in energy usage does not necessarily mean that it falls into the category of energy theft. There is a possibility that there might be a reduction in energy usage because of any technical issue. So, the proposed *GrAb* model considers all three possible scenarios, technical loss, non-technical loss (energy theft), and normal consumption of energy, which will be discussed in detail in the subsequent section.

## 3. System Model and Problem Formulation

The energy consumption prediction and its profiling are some of the significant aspects to identify the theft of energy using a data-driven approach to characterize the behavior of consumers. These approaches identify energy usage deviations accurately, i.e., possible energy theft. This section presents a proposed *GrAb* system model and problem formulation for energy theft detection.

### 3.1. System Model

Figure 3 depicts the system model of the proposed *GrAb* system to carry out the theft detection based on the energy consumption prediction followed by the classification of the energy loss profile. The *GrAb* system is divided into two stages: (1) energy consumption prediction and threshold calculator and (2) classification of energy loss. The energy consumption data are fetched from the SMs of residences. These collected energy data $L_{d}$ can be inconsistent in nature, such as missing values, incorrect values, and having a varying range of values, which requires pre-processing. It is pre-processed by filling the missing values using a central tendency, i.e., mean values. Next, to solve the range problem of the dataset load values ($L_{d}$ values), we applied z-score normalization and linear interpolation to scale the values appropriately, enhancing the LSTM model’s accuracy. Furthermore, the proposed model referred to the time-series-based hourly energy consumption data, where data values are closely spaced. The openEI dataset is considered to train the LSTM model for energy consumption data prediction with the help of load data. The dataset we referred to is a standard dataset that has very few missing values; thus, employing linear interpolation can easily identify these missing values and fill them appropriately. We have explored various other techniques such as finding the mean, median, etc., but linear interpolation has given better results. In case of a long period of missing data or even peaks, the multiple imputations could be applied by using statistical or machine learning-based techniques. Furthermore, incorrect load values $L_{d}$ are corrected using the internal functions of pandas and numpy, such as fillna(), isna().sum(), dropna(), and many more. It furnished load consumption value $L_{d}$ for the *i*th hour ($h_{i}$) of the day as $\omega^{h_{i}}$ and then passed it to the threshold calculator and LSTM prediction model individually.

In stage 1, the LSTM–based prediction model is presented to predict the hourly load values based on the historical data of load consumption $\omega^{h_{i}}$. The LSTM model consists of three layers, which include one input layer, one middle layer, and one dense/output layer. The input and middle layers use two dropout layers to combat overfitting in the model. The hourly predicted load values Pred represent the estimated load consumption at hour *i*, in accordance with the LSTM prediction model. Parallelly, in stage 1, the energy threshold calculator gives the threshold limit for energy consumption in an hour *i*th. This helps in classifying the energy loss, i.e., TL or NTL or normal profile (NP). If the obtained load values deviate within this limit, the load consumption is considered NP; otherwise, it is classified as TL or NTL. The threshold calculator will assign the threshold energy value for each hour *i* of the day $day_{d}$, which will be calculated using the total ψ consumed in hour *i* throughout the month in which $day_{d}$ falls. The threshold calculator will correspondingly give the hourly threshold consumption value α. Both Pred and α are passed into the SVC-based loss classifier.

In stage 2, The SVCM-based linear energy loss classifier has been designed to assign label $SV_{l}$ for energy loss classification (to display a higher or lower energy consumption) for each hour of the day. Therefore, every hour *i* of the day is assigned a label $SV_{l}$ according to the consumption done. The total count of all these labels is used to decide energy theft for a specific day $day_{d}$. In case energy theft is not detected, the *GrAb* determines if there has been any TL or it has been normal consumption throughout the day using the same count of labels $SV_{l}$.

### 3.2. Problem Formulation

As discussed in the above subsection, energy ψ is transmitted and distributed by various sources *S*. Each of these sources provides energy to consumers $C_{j}$, who, in this case, are considered residential energy consumers. Each consumer $C_{j}$ is making use of several appliances {$A_{1}$, $A_{2}$, ⋯, $A_{m}$} ∈A. The energy consumption for each of these devices used by consumer $C_{j}$ may fall into an energy loss type represented by class {$\gamma_{1}$, $\gamma_{2}$, $\gamma_{3}$} ∈γ referred to as technical loss, non-technical loss and normal consumption. The total consumption of a day $day_{d}$ is defined as follows.
(1)$$\psi_{day_{d}}=\sum_{i=1,y=1}^{i=24,y=3}\psi_{day_{d}}^{{\left(h_{i}\right)}_{\gamma_{y}}}.$$
where $h_{i}$ represents the *i*th hour and $\gamma_{y}$ shows the energy loss type ∈ {$\gamma_{1}$, $\gamma_{2}$, $\gamma_{3}$} of a particular day $day_{d}$. The energy consumption prediction on the hourly basis $\omega^{h_{i}}$ is produced while passing the hourly load value, i.e., $\psi^{h_{i}}$ to the LSTM model of *GrAb*.
(2)$$\omega^{h_{i}}=LSTM\left(\psi^{h_{i}}\right)$$

The threshold hourly energy consumption for an hour $h_{i}$ in a month $M_{x}$ is defined by the entity α as follows.
(3)$$\alpha_{M_{x}}^{h_{i}}=\left(\sum_{d=1}^{r}\sum_{i=1}^{24}\psi_{day_{d}}^{h_{i}}\right)/r$$
where *r* is the number of days in the month $M_{x}$ which may vary from 28 to 31 while $day_{d}$ is the *d*th day of the month and $h_{i}$ is the *i*th hour of that day. Let us consider Δ as the absolute difference function and μ is the meter readings recorded by the user for day $day_{d}$. Let the SVC labels be {$SV_{1}$,$SV_{2}$} contained in set SV be defined as 1 and −1, respectively. Let θ(a,b) be the comparator function for two values *a* and *b*. The functioning of θ(a,b) can be defined as follows.
(4)θ(a,b)=1,∀a≥b
and
(5)θ(a,b)=0,∀a<b

Now, let ξ be the threshold limiter.
(6)$$\xi =\theta \left(\Delta \left(\mu_{h_{i}},\omega_{h_{i}}\right)\right),\Delta \left(\alpha_{h_{i}},\omega_{h_{i}}\right)).$$

Then, label $SV_{l}$ is defined as follows.
(7)$$SV_{l}=1,\forall \xi =1,\mu_{h_{i}}-\omega_{h_{i}}>0$$
and
(8)$$SV_{l}=-1,\forall \xi =1,\mu_{h_{i}}-\omega_{h_{i}}<0$$

For each hour $h_{i}$ of the day, a label $SV_{l}$ is given for that hour, considering the parameters and operations shown in the above equation. The problem formulation for *GrAb* comes down to identify the type of energy loss $\gamma_{y}$ occurred in day $day_{d}$ as follows.
(9)$$\gamma_{y}^{day_{d}}=\Phi \left(\sum_{i=1}^{24}SV_{1},\sum_{i=1}^{24}SV_{2}\right)$$
where Φ is the loss classification function, subject to the constraints:(10)$$\forall \gamma_{1},\sum_{i=1}^{24}SV_{1}\geq \sum_{i=1}^{24}SV_{2}$$
(11)$$\forall \gamma_{2},\sum_{i=1}^{24}SV_{1}<\sum_{i=1}^{24}SV_{2}$$
(12)$$\forall \gamma_{3},\sum_{i=1}^{24}SV_{1}=0,\sum_{i=1}^{24}SV_{2}=0$$
(13)$$\forall SV_{1},\mu_{h_{i}}>\omega_{h_{i}}$$
(14)$$\forall SV_{2},\mu_{h_{i}}<\omega_{h_{i}}$$
(15)$$\mu_{h_{i}},\omega_{h_{i}},\alpha_{h_{i}}\geq 0$$

## 4. *GrAb*: The Proposed Approach

Figure 4 depicts the architecture of the DL-based energy theft detection system referred to as *GrAb* to ensure the reduction in non-technical energy loss. *GrAb* consists of four layers: (i) energy data generation and collection layer (EDGCL), (ii) threshold calculator layer (TCL), (iii) energy consumption prediction layer (ECPL), and (iv) energy theft detection layer (ETDL). Each layer is described as follows.

### 4.1. Energy Data Generation and Collection Layer

In this section, the energy plays a key role in the working of the electric appliance ($A_{k}$), which needs a larger amount of energy generation (at SG) for consumers {$C_{1}$, $C_{2}$, ⋯, $C_{n}$} ∈C to operate their electric appliances {$A_{1}$, $A_{2}$, ⋯, $A_{m}$} ∈A at time T. The smart meter (SM) records the energy consumption data generated by various electrical appliances such as AC, refrigerators, computers, television, heater, lights, fans, and many more are recorded by the smart meter (SM). Here, SM collects the energy consumption data on an hourly basis for each appliance $A_{k}$. Energy generation sources fulfill the energy demands of these electric household appliances to carry out day-to-day activities. There are various sources of energy such as wind power, hydropower, thermal power and many more {$S_{1}$, $S_{2}$, ⋯, $S_{o}$} ∈S, which provide energy to consumers {$C_{1}$, $C_{2}$, ⋯, $C_{n}$} ∈C, who belong to different domains {$D_{1}$, $D_{2}$, $D_{3}$} ∈D such as residential houses, industries and the commercial sector. The various amount of energy {$\psi_{1}$, $\psi_{2}$, ⋯, $\psi_{q}$} ∈ψ generated by the *u*th source $S_{u}$ at SG are transmitted to the relevant transmission stations and then distributed to every domain {$D_{1}$, $D_{2}$, $D_{3}$}. In the case of the residential houses, $D_{1}$, the energy consumed by consumer $C_{j}$ using appliance $A_{k}$ is recorded through SM. In the *GrAb* model, we have considered the hourly energy consumption data for the prediction of energy theft—the $\psi_{q}$ amount of energy data generated—and collected the required pre-processing for energy theft detection.

### 4.2. Threshold Calculator Layer

In this layer, the energy data obtained from the SM are pre-processed and converted to pre-processed data $D_{p}$. Algorithm 1 shows a brief pseudocode of this layer. This $D_{p}$ includes data normalization between 0 and 1 to obtain better computational results. To deal with the potential missing and inconsistent values, the *GrAb* model uses the linear interpolation technique LIT to remove hidden inconsistencies from the collected energy data. The $D_{P}$ contains energy load consumption in the *i*th hour, $\omega^{h_{i}}$, which plays a crucial role in threshold calculation and energy consumption prediction. Here, $\omega^{h_{i}}$ is passed to a threshold calculator to calculate the hourly threshold value of energy consumption for a specific day. $C_{thres}$ is a decision-maker to identify energy theft in the final stage. This $\omega^{h_{i}}$ for each appliance $A_{k}$ is passed to the LSTM prediction model for energy consumption prediction, which is discussed in detail in the subsequent section.
**Algorithm 1:** Step-by-step process to calculate the threshold value**Input**: Month $M_{x}$, Energy consumption values $\omega^{h_{i}}$**Output**: Average Consumption $C_{avg}$**Initialization**: Let *d* be the no. of days in month $M_{x}$, and $h_{i}\in$ be set of hours {$h_{1}$, $h_{2}$, ⋯, $h_{24}$}. Let $C_{avg}\left[h_{i}\right]$ define the threshold consumption at $h_{i}$ th hour.   **procedure**CALCULATE_HOURLY_THRESHOLD($\omega^{h_{i}}$, $M_{x}$)       **for** i← 1 to *d* **do**           **for** j← 1 to 24 **do**              $C_{avg}\left[h_{i}\right]\leftarrow$$C_{avg}\left[h_{i}\right]$ + $\omega^{h_{i}}$           **end for**       **end for**       **for** i← 1 to *d* **do**           **for** j← 1 to 24 **do**              $C_{avg}\left[h_{i}\right]\leftarrow$$C_{avg}\left[h_{i}\right]÷d$           **end for**       **end for** return $C_{avg}$**end procedure**

### 4.3. Energy Consumption Prediction Layer

In this layer, the $D_{p}$ data contain a time-series data $\omega^{h_{i}}$, which is passed to the LSTM-based prediction model of *GrAb*. This LSTM model comprises two LSTM layers, two dropout layers, and one dense layer. The LSTM model produces predicted load values Pred on an hourly basis for each appliance $A_{k}$. In the proposed LSTM model, the overfitting of the time series energy data is handled by including the dropout after layer-1 and layer-2 in the *GrAb* system, with 50 nodes in each layer. The proposed model is a sequential model with a dropout of 5% for its LSTM layers. Dropout is included with each LSTM layer as they have a greater number of nodes and parameters and are thus more prone to cause overfitting, as shown in Algorithm 2. Table 3 summarizes the details of the proposed LSTM-based energy consumption prediction model. Then, the dense layer acts as an output layer for forwarding the predicted energy load data Pred to the next layer, i.e., the energy theft detection layer.
**Algorithm 2:** Working of the LSTM-based proposed model**Input**: Dataset DS, Dropout dOut**Output**: Predicted load values Pred**Initialization**: From DS Consider list of *V* load values, {$L_{1},L_{2},\cdots ,L_{V}$} for the *k* number of appliances {$A_{1}$, $A_{2}$, …, $A_{k}$} ∈ANumber of nodes in LSTM model is = $N_{LSTM}$1:**procedure**LSTM_LOAD_FORECASTING( DS, dOut)2:    Identify Features from DS3:    **for** v← 1 to V **do**4:        L[v]← INPUT()5:    **end for**6:    **for** v← 1 to V **do**7:        Normalize L[v] in range (0,1)8:    **end for**9:    **for** v← 1 to V **do**10:        **if** L[v].IsNotValid **then**11:           L[v]←L[v] Interpolate()12:        **end if**13:    **end for**14:    $I_{seq}\leftarrow$ INITIALIZE_SEQUENTIAL_LSTM_ MODEL(*DS*)15:    $LSTM_{Layer_{1}}\leftarrow$ INPUT($L_{v}$)16:    $LSTM_{Layer_{1}}$.DROPOUT(dOut)17:    $LSTM_{Layer_{2}}\leftarrow$ COMPUTE($LSTM_{Layer_{1}}$)18:    $LSTM_{Layer_{2}}$.DROPOUT(dOut)19:    $Dense_{Layer}\leftarrow$ COMPUTE($LSTM_{Layer_{2}}$)20:    Pred← PREDICT_LOAD_VALUES($Dense_{Layer}$)21:    return Pred22:**end procedure**

### 4.4. Energy Theft Detection Layer

In this layer, the most necessary part of the classification of the predicted energy profile takes place. The predicted load values Pred and threshold values $C_{thres}$ are considered in this layer for the classification. Along with them, we also consider the meter readings $SM_{R}$, which are the real-time energy consumption data values given by the SM. All of these values are considered on an hourly basis, and the accumulated results of each hour are considered to declare the type of consumption that has happened throughout the day. We proposed a SVM-based linear loss classifier, i.e., SVC for computational analysis of $SM_{R}$, Pred and $C_{thres}$, as shown in the loss classifier Algorithms 3 and 4. Here, the energy consumed throughout the day is considered energy theft or NTL (ET) if the energy consumption prediction Pred is relatively lower than the $C_{thres}$ value. In the case of no ET, the *GrAb* system checks for the scope of any TL of the energy consumption of the specific day. If no ET and no TL exist, then the energy consumption throughout the specific day is declared to be normal consumption NR.
**Algorithm 3:** SVC-based class labeling**Input**: Test data value *z***Output**: Class Label $C_{l}\in \gamma_{y}$**Initialization**: Sample Training values as X[] and Sample Training Labels as Y[]1:**procedure**SVM_CLASSIFICATION( *z*, $\omega^{h_{i}}$)2:    clf← INITIALIZE_SVC_ MODEL()3:    SET(Kernel←linear )4:    SET(Gamma←auto)5:    clfFIT(X[],Y[])6:    $C_{l}\leftarrow$clf.PREDICT(*z*)7:    return $C_{l}$8:**end procedure**

**Algorithm 4:** Loss classifier of the proposed work
**Input**: Smart meter readings $SM_{R}$, Predicted load values Pred, Average hourly load consumption $C_{avg}$, Day $day_{d}$
**Output**: Usage classification State St
**Initialization**: Let w[] record if the consumption has been higher than threshold on an hourly basis. Let z[] record if the consumption has been lower than the threshold on an hourly basis. Let $high_{count}$ be the count of upper limit of threshold crossed in the day, and let $low_{count}$ be the count of lower limit of threshold crossed in the day
1:**procedure**MAKE_CLASSIFICATION( $T_{x}$, BC)2:    **for** i← 1 to 24 **do**3:        **if** MOD($SM_{R}\left[day_{d}\right]\left[h_{i}\right]$-$Pred\left[day_{d}\right]\left[h_{i}\right]$) >4:             MOD($C_{avg}\left[day_{d}\right]\left[h_{i}\right]$-$Pred\left[day_{d}\right]\left[h_{i}\right]$) **then**5:            $z\leftarrow SM_{R}\left[day_{d}\right]\left[h_{i}\right]-Pred\left[day_{d}\right]\left[h_{i}\right]$6:            res←SVM(z)7:            **if** res= 1 **then**8:                w[i]←19:                **if** res= −1 **then**10:                    z[i]←111:                **end if**12:            **else**            continue13:            **end if**14:         **end if**15:    **end for**16:    **for** i← 1 to 24 **do**17:        **if** w[i]=1 **then**18:           $high_{count}\leftarrow$$high_{count}$ +119:        **else**20:           **if** z[i]=1 **then**21:               $low_{count}\leftarrow$$low_{count}$ +122:           **else**            continue23:           **end if**24:        **end if**25:    **end for**26:    **if** $high_{count}\geq low_{count}$ && $high_{count}>0$ **then**27:        $St\leftarrow ``TECHNICAL\;LOSS^{\prime \prime }$28:        **if** $low_{count}>high_{count}$ **then**29:           $St\leftarrow ``THEFT\;DETECTED^{\prime \prime }$30:        **else**31:           $St\leftarrow ``NORMAL^{\prime \prime }$32:        **end if**33:    **end if** return St34:
**end procedure**



## 5. Results and Discussion

The *GrAb* model works in two stages; as discussed in Section 3, stage 1 comprises the energy consumption prediction and threshold calculation, and stage 2 classifies the energy loss for energy theft detection. In this section, we have emphasized the discussion on results obtained from both the stages, i.e., results obtained from the proposed LSTM-based prediction model and classification of energy loss. The proposed *GrAb* model is used over an openEI dataset that consists of hourly consumption data.

### 5.1. Dataset Description

The proposed *GrAb* model is used over a standard benchmarked dataset, i.e., Open Energy Information (OpenEI) dataset that provides free and open-source real-time energy data for researchers, energy traders, policymakers, and technology enthusiasts to extract useful information and analysis and provide the effective decisions in SG systems. The dataset consists of hourly consumption data of several electrical devices such as AC, Light, Basic Facilities, and Miscellaneous (Misc) (considered for the experiment) for a residential house located in Alaska [44]. Furthermore, it has a time series column that specifies the date and time the hourly consumption data of electrical appliances are taken. Because of the inclusion of time series data, we have incorporated the LSTM model, which works efficiently with time series data. The dataset comprises 8761 × 4 rows and columns of hourly consumption data, which passes as an input to the AI-based LSTM model for energy consumption prediction. However, the dataset is not appropriate for training because of missing values, sensitivity to outliers, and not normalized. Therefore, appropriate python libraries are used, such as pandas and NumPy, which have default functions such as fillna(), isna(), sum, and dropna that help to confront missing values. Furthermore, the interpolation technique has been utilized to solve the value range problem, i.e., in a single column, a few values are higher (6.22669 kW), and a few values are smaller (0.96790 kW); this degrades the performance of LSTM training. Therefore, the interpolation technique seamlessly standardized the value of all column values to a specific range, i.e., [—1, 1]. Then, the pre-processed data are forwarded to the LSTM model to be trained on a time series dataset of hourly consumption data, i.e., OpenEI.

### 5.2. Simulation Settings

The proposed *GrAb* model is implemented on a Windows operating system (OS) configured as Intel(R) Core(TM) CPU @ 2.60 GHz, 16 GB RAM using python, i.e., a functional programming language. Next, various open Source libraries such as Numpy v1.18.4, Pandas v1.0.4, and Keras v2.3.1 have been used to perform miscellaneous computations of DL libraries. The LSTM model learns the dynamic trends of energy consumption data for a particular residential customer.

### 5.3. Performance Evaluation and Comparative Analysis of LSTM Model

The energy load values predicted by the LSTM model will act as a reference to compare with the consumption data. Therefore, we have considered the energy load predictions as our actual curve. The OpenEI dataset is employed for the training and testing purposes, which is divided into a 70:30 ratio to get satisfying accuracy in prediction results. Here, Root Mean Squared Error (RMSE) and mean absolute percentage error (MAPE) are considered as the evaluation parameter for the proposed LSTM-based prediction model, as shown in Figure 5. LSTM is formally used as a time forecasting training model; hence, to evaluate the forecast accuracy of the LSTM model, RMSE and MAPE are commonly used as performance metrics. MAPE is defined as the average percentage error between the ground truth value and a forecast value for each time period. In our work, predicted energy load data Pred is the output of the LSTM model; it can be formulated in MAPE as
(16)$$\mathrm{MAPE}=\frac{1}{n}\;\sum_{t=1}^{n}\frac{|\hat{Pred}-Pred|}{Pred}$$
where, *n* represents all the data points for each time *t*, Pred indicates ground truth value and $\hat{Pred}$ specifies the forecast predicted value. Contrary, the RMSE is defined as the square root of the difference between the predicted value and ground truth value for the energy load data Pred, it is formulated as
(17)$$\mathrm{RMSE}=\sqrt{\sum_{t=1}^{n}\frac{{\left(\hat{Pred}-Pred\right)}^{2}}{n}}$$

The lower the MAPE and RMSE error values, the better the performance of prediction by the LSTM model. Figure 5 illustrates the RMSE values predicted by the LSTM model for different electrical appliances. The higher the RMSE values, the lower the model’s prediction. From the graph, it is evident that the LSTM has shown an accurate prediction for miscellaneous and AC, as it has a minimum error (0.005 and 0.0010). Contrary, the LSTM model underperforms while predicting basic facilities and light load consumption as it has higher RMSE values, i.e., (0.025 and 0.015). It is important that the LSTM model correctly predicts the load consumption data of the electric appliances that are regularly used and consumes more energy, such as AC, light, fan, and others. This is because such appliances consume more energy and produce high-energy tariff bills for the consumers. Therefore, the forgers try to modify the load consumption data of such appliances to get the minimum tariff bill. Needless to say, the proposed work has accurately predicted the load consumption data of the above-mentioned electric appliances, which results in reduced energy theft in smart grids. Table 4 shows a comparative analysis of the prediction error of MAPE and RMSE for different electrical appliances. The higher the prediction error, the lower the LSTM model’s performance. From Table 4, we can observe that the MAPE prediction error is higher than the RMSE prediction error; this happens because it calculates the absolute average percentage error, but RMSE uses square root to minimize the error. In addition, MAPE is not considered a valuable performance metric when dealing with the time series dataset because MAPE is not differentiable, and its hessian is zero wherever it is defined. So, there might be a need for differentiation when forecasting future energy consumption values. Therefore, RMSE is a reliable performance metric as it is differentiable at any instant of time. The error values show that the miscellaneous has a lower RMSE value and basic facilities have a higher RMSE value (also shown in Figure 5). Incorporating RMSE makes the model highly accurate in load consumption prediction against the other exiting approaches. The last row shows an average load consumption prediction error for both MAPE and RMSE; the value of RMSE is small, indicating that the predicted and actual data are close to each other, which implies that the model has better accuracy. So, we consider the results of the prediction (using the LSTM model) as a basis on which the load classification is employed in the *GrAb* model.

The efficiency of the GrAb LSTM prediction model is being compared to the other state-of-the-art approaches that also use the LSTM-based load forecasting model for energy consumption data of residential houses [45]. Khan et al. [46] (Baseline 1) presented an LSTM-Autoencoder-oriented DL approach to predict the load consumption at the household level by using the UCI machine learning repository [47]. This approach comes up with the prediction results with a low error rate, i.e., an RMSE value of 0.47.

In contrast, the *GrAb* LSTM model gives an RMSE value of 0.0137 (considering the average RMSE value from Table 4), which is quite low, as shown in Figure 6a. To validate the accuracy of the *GrAb* LSTM model, we have obtained MAPE as one of the parameters to compare with state-of-the-art approaches. Choi et al. [48] (Baseline 2) presented an LSTM-based mixed data sampling (MIDAS) model to predict the load consumption for short and long term on a residential dataset [49]. The MIDAS produces a value of 32.5 MAPE, where the *GrAb* produces a 7.42 average MAPE value, which outshines the *GrAb* in terms of MAPE as well compared to other state-of-the-art approaches, as shown in Figure 6b.

### 5.4. Energy Loss Classification Results

As the dataset mentioned in the previous subsection does not have any traces of theft data, we injected synthetic data to simulate the theft scenario for classification of energy losses and theft detection using the *GrAb* model. The false data injection is accomplished by analyzing the probability distribution of the dataset because once the synthetic generator generates the data, it should map the probability distribution of the original dataset. However, we want a forge data to be inserted into the dataset to efficiently train our AI models for the prediction and detection of energy theft. Hence, the synthetically generated data will not entirely follow the probability distribution of the original dataset. To do so, we have used a conditional generative adversarial network (CTGAN) by employing an open-source python library, i.e., CTGAN. First, we have imported an essential library of CTGAN, i.e., CTGANSynthesizer, wherein we have provided hyperparameters, such as batch_size, epochs, and verbose to initialize the model training. Then, with the help of the fit() function, we trained the model to generate 1500 synthetic data samples, which are then appended to the original dataset. The principle for this synthetic data injection is that a counterfeit customer can tamper with the home energy system (by tampering the smart meter or other means) to reduce the energy consumption value reported to the SG. For this purpose, we injected false data values for each appliance for a specific customer and compared them with the consumption data calculated based on the threshold values of the genuine customer. We have considered both the positive and negative deviation of load consumption to detect such technical loss.

The classification result of energy loss for each scenario is summarized in Table 5, and Figure 7 shows the hourly distribution of positive, negative, and neutral deviation of energy consumption for each appliance.

#### 5.4.1. Case I—Technical Loss

Suppose the injected data hourly consumption value is less than the threshold value; then, it is considered energy theft. If it is more than the threshold value, it could be a technical loss, and the rest falls as the normal load consumption profile. For example, let us take the traces of 14th May data for appliance AC and inject the false data for the given day. Then, by using the LSTM model, we have predicted the values of the load consumption and compared them with the actual (baseline values marked in green color) values, as shown in Figure 8.

Due to the high accuracy of the proposed LSTM model, we consider the model’s predictions as our actual curve to compare with the anomalous data for theft detection. As shown in Figure 8, the anomalous hourly consumption is marked in red color and shows the technical energy losses. Then, the normal load profile is marked in sky-blue color.

For the given scenario of Figure 8, the load consumption Pred at 3 a.m. is 0.1312 kWh. The average consumption for an hour at 3 a.m. during May is 0.1486 kWh. Therefore, a difference of 0.017k Wh is valid to consider the consumption normal. Now, looking at the anomalous data, the consumption at 3 a.m. shown by the meter reading is 0.1046 kWh. The magnitude of its deviation from the baseline curve is 0.026 kWh. Since this difference in consumption is greater than the one allowed for consumption at 3 a.m., the SVM-based loss classifier algorithm classifies it as a reading lower than allowed and updates L[3]=1.

Conventionally, for any particular hour *i*, in case the magnitude of difference between predicted load consumption (actual) and baseline value is greater than that of the threshold value, the load consumption is verified if they fall on the higher or lower end, and correspondingly, the list H[] and L[] are updated, respectively, for an hour *i* to detect technical energy losses. Similarly, technical energy losses for other appliances such as light, Misc, and Basic facilities have been reported.

The LSTM model is used to predict the value of load consumption (normal load profile is marked in sky-blue color) and compare it with baseline values (green line), as shown in Figure 9. The graph shows the energy loss prediction for an electric appliance, i.e., light. Furthermore, it depicts the normal energy loss prediction where the baseline (green line) shows the ground truth values and the red line shows the predicted anomalous load profile values of the LSTM model. Figure 9 shows the abnormal behavior in the LSTM-based predicted load profile values; this happens because we have synthetically injected the data (to detect the energy theft), which is not in the distribution of the original load profile data. From the graph shown in Figure 9, it can be observed that at a time (t = 11), the baseline shows 0.064 kWh load consumption; however, the anomalous load shows 0.13 kWh load consumption. Similarly, at a time (t = 21), we can notice the same malicious behavior of the forger. When the actual value fluctuates from the baseline value, there is a high possibility of an energy theft scenario.

#### 5.4.2. Case II—Energy Theft

We have injected synthetic data to create a scenario of energy theft in which malicious customers have lowered their meter readings for unethical means. The malicious user can lower his own appliances load and might increase the load values of his neighborhood so that the overall energy consumption of the locality remains the same. Thus, no one might be able to detect his meter tampering practice.

To handle the aforementioned issue, the threshold energy consumption value is calculated in the proposed *Grab* model to identify energy theft, i.e., energy consumption < threshold value. Here, *GrAb* analyzes each resident’s appliance consumption individually to detect energy theft on a practical basis to grab the unethical practices of the malicious customer.

Here, the load consumption analysis for energy theft of the basic facility is shown in Figure 10. It illustrates the normal load profile (marked in sky-blue color) of basic electric appliances, where the baseline (green line) indicates the existing values and the anomalous load profile (red line) indicates the predicted values from the LSTM model. In the figure, both sky-blue and green lines overlap, which signifies that there is no data integrity issue in the load consumption data, i.e., no energy theft. However, it is noticeable from the graph (Figure 10) that the load profile of the basic electric appliances deviates from its normal average load profile, indicating an energy theft. At the time (t = 7), the baseline load consumption and normal value are 1.2; however, due to the false injection, the anomalous value fluctuates from the baseline value, i.e., the value is 0.9. The same is also noticeable at the time (t = 20); the baseline load consumption value is 1.7; however, the actual predicted value is 1.5, indicating the case of energy theft.

Furthermore, Figure 11 depicts the energy loss prediction for miscellaneous electrical appliances. It shows the normal load consumption profile (sky-blue line) for miscellaneous electrical devices for a specific day. In contrast, the graph shows the anomalous load profile (red line) for the same day to notice the energy theft. The baseline value (green line) indicates the ground truth values, and the anomalous value (red line) indicates the predicted values by the LSTM model. From Figure 11, we can observe that the baseline and normal loads are overlapped with each other; this happens because the LSTM model correctly predicts the load profile for the electrical device. However, the graph shows the malicious behavior of the attacker, where it has injected a false reading in the load profile. For instance, at a time (t = 20), the baseline value is 0.61; however, the anomalous value has fluctuated to 0.53, which signifies a scenario of energy theft.

#### 5.4.3. Case III—Normal Energy Consumption

This case comprises the magnitude of difference between the predicted consumption of the synthetic data and the baseline values in a specific hour. If this difference is in the range of the allowed fluctuation calculated from the threshold calculator of the *GrAb* model for the specific appliances within the particular hour, such a case can not be considered as any loss, and it is normal energy consumption.

## 6. Conclusions

The modernization of grid infrastructure, i.e., SG, has improved energy generation, distribution, and management. Even though the adoption of SG is not yet uniform around the world, many countries have started investing heavily in SG deployment. However, energy management becomes difficult due to energy losses either via technical or non-technical reasons (i.e., energy theft), which increases the energy generation burden on SG. In this paper, we propose a data-driven analytical model referred to as *GrAb* to identify the energy theft in a residential area. The *GrAb* used the DL-based LSTM model to make predictions on energy consumption data and proposed a threshold calculator. Then, an SVM-based loss classifier is used to classify energy consumption into three classes—technical, non-technical, and normal energy consumption—based on the threshold value. Implementing the proposed model in a real-time environment can benefit society and the government agencies by catching the people who resort to unethical practices such as energy theft. Furthermore, the *GrAb* also benefits the resident owners by ensuring that they are not using faulty smart meters and helping genuine customers reduce the excess energy bill caused due to energy theft. In the future, the *GrAb* can be extended to commercial areas and industries to increase the revenues of the utility providers. Additionally, we will employ AI-based federated learning and integrate it into the smart meters of the smart grid, which will detect the energy theft in the local smart meter of a particular individual as well as the global smart meter in case the attackers have wirelessly maneuvered the smart meter of a large geographic area.

## Figures and Tables

**Figure 1 sensors-22-04048-f001:**
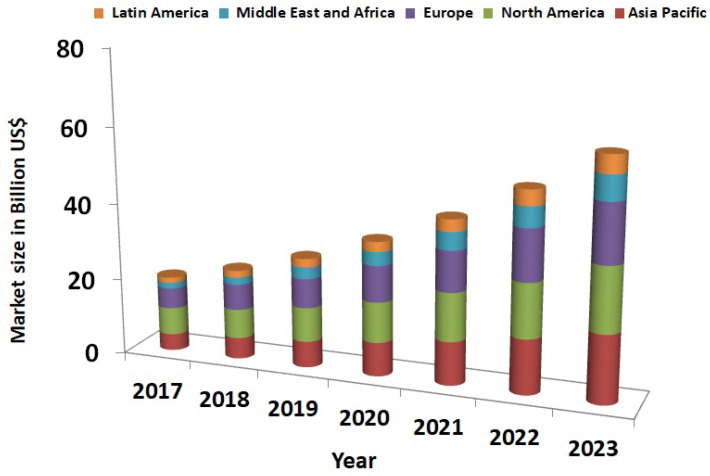
Smart Grid Market Growth [6].

**Figure 2 sensors-22-04048-f002:**
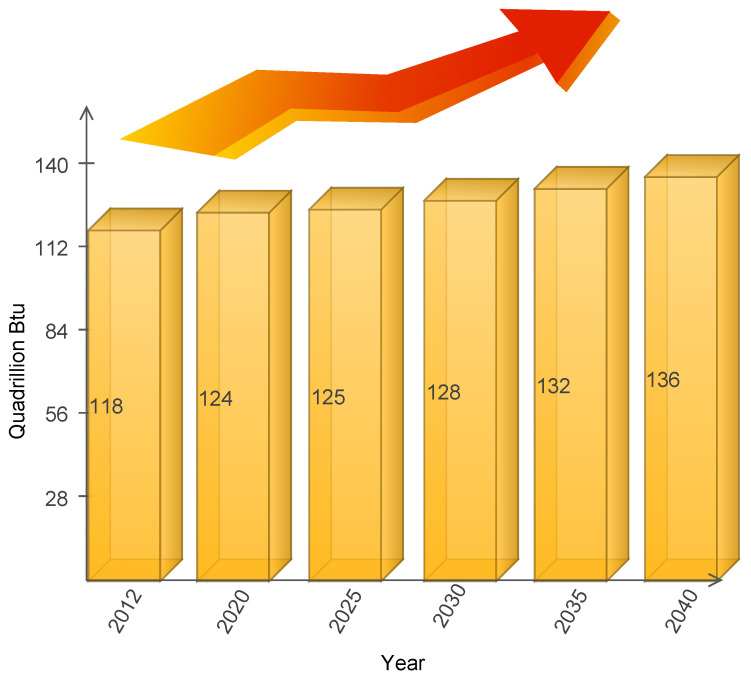
Energy Growth Forecasting Using Deep Learning.

**Figure 3 sensors-22-04048-f003:**
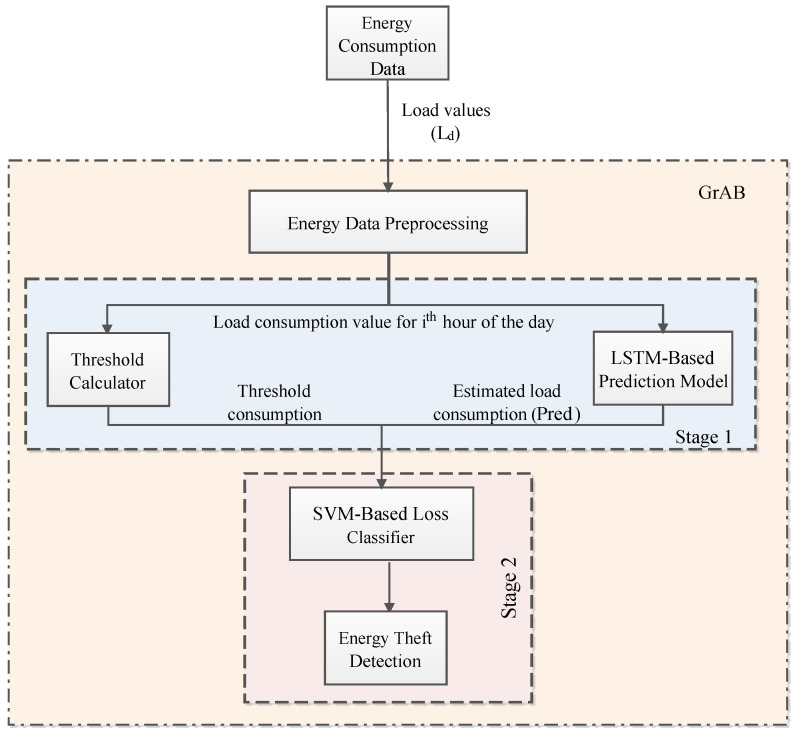
*GrAb* System Model.

**Figure 4 sensors-22-04048-f004:**
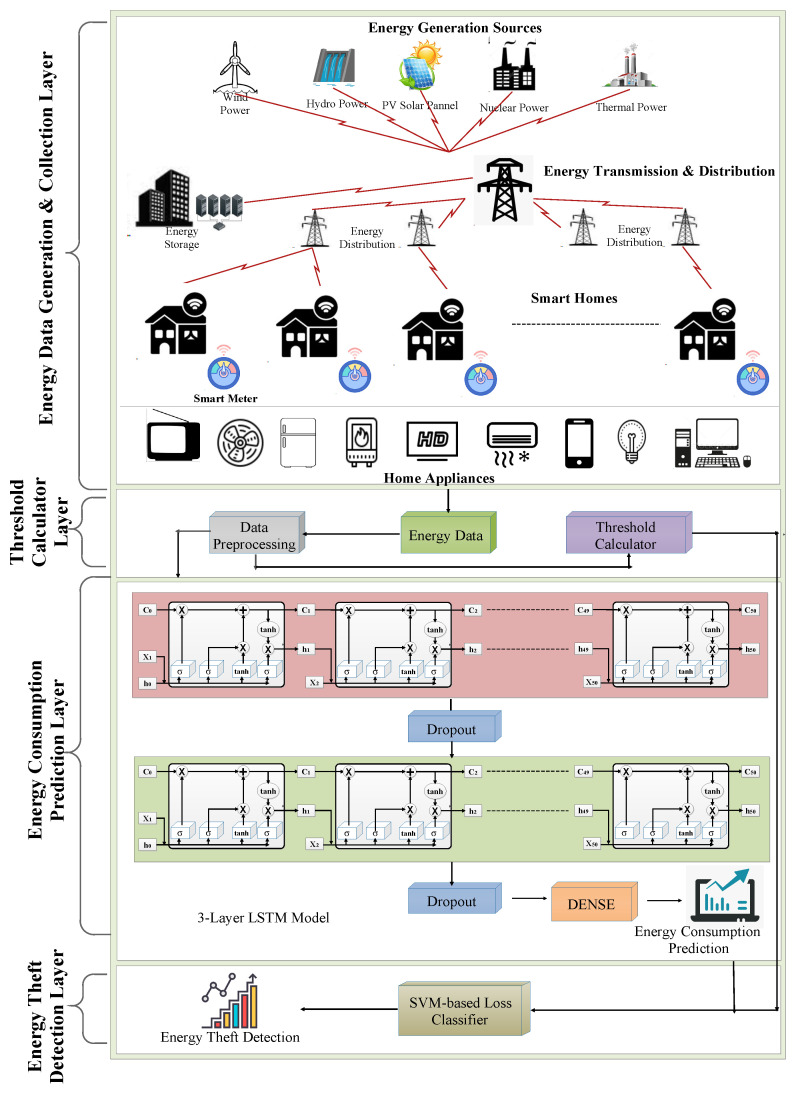
*GrAb* system architecture.

**Figure 5 sensors-22-04048-f005:**
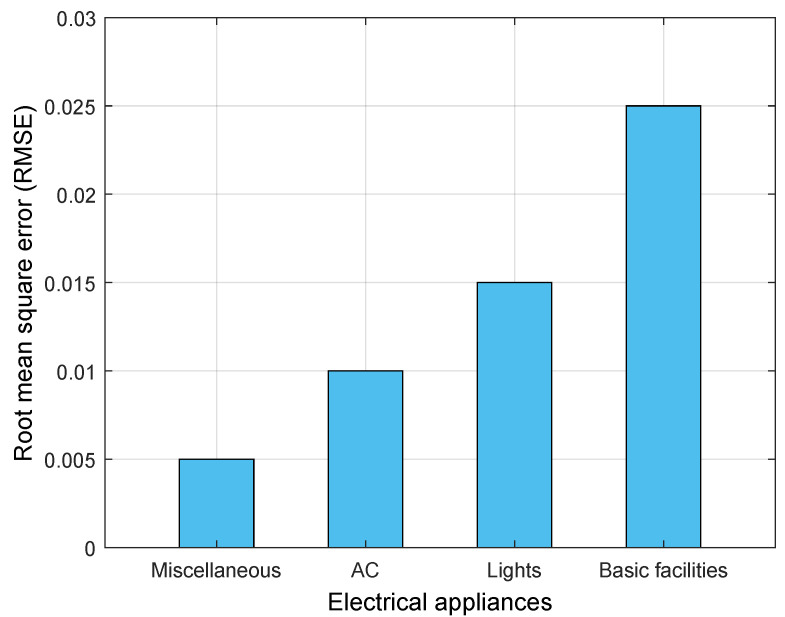
RMSE values for devices using *GrAb*.

**Figure 6 sensors-22-04048-f006:**
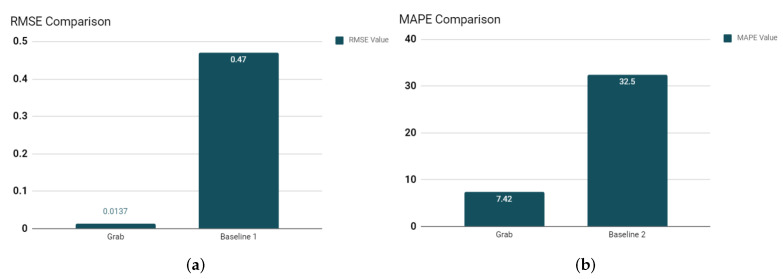
Comparison of the proposed *GrAb* model and state-of-art approaches. (**a**) *Grab*—RMSE Comparison. (**b**) *Grab*—MAPE Comparison.

**Figure 7 sensors-22-04048-f007:**
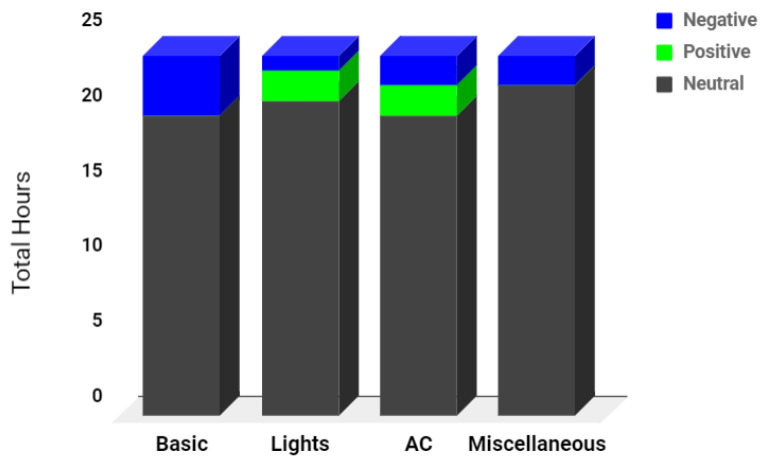
Distribution of hours for deviation.

**Figure 8 sensors-22-04048-f008:**
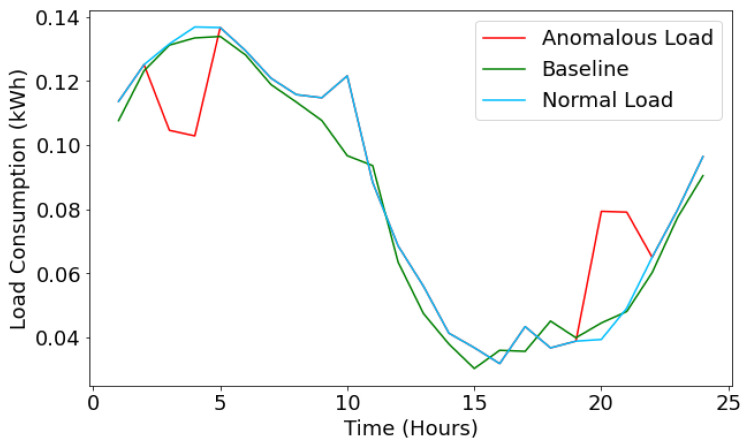
Energy loss prediction for AC.

**Figure 9 sensors-22-04048-f009:**
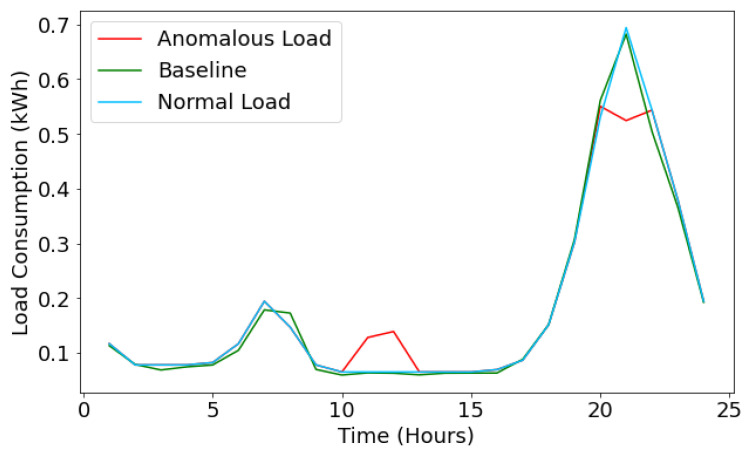
Energy loss prediction for lights.

**Figure 10 sensors-22-04048-f010:**
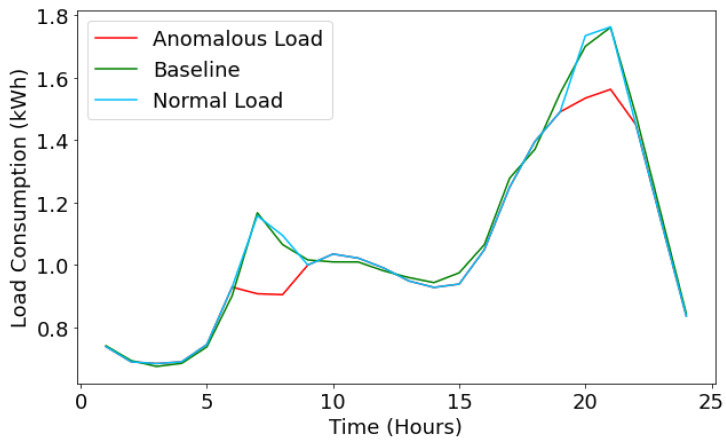
Energy Loss prediction for Basic Facilities.

**Figure 11 sensors-22-04048-f011:**
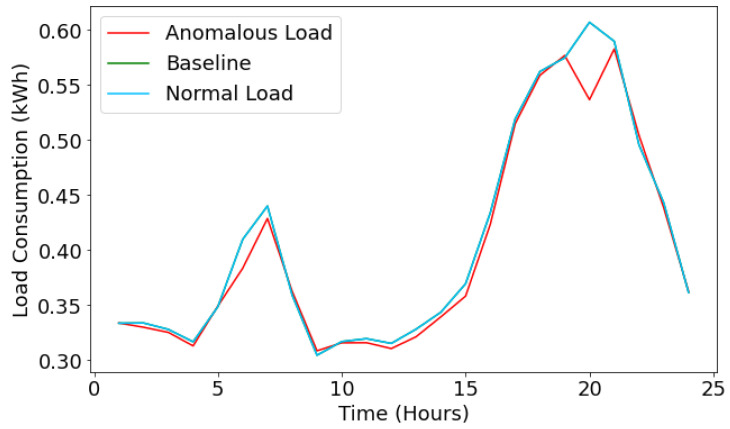
Energy loss prediction for miscellaneous devices.

**Table 1 sensors-22-04048-t001:** Abbreviations Used.

Symbols	Description	Symbols	Description
AI	Artificial Intelligence	ICT	Information Communication Technology
SG	Smart Grids	EPRI	Electric Power Research Institute
LSTM	Long Short-Term Memory	RFM	Random Forest Model
RMSE	Root Mean Square Error	EDGCL	Energy Data Generation and Collection Layer
MAPE	Mean Absolute Percentage Error	TCL	Threshold Calculator Layer
St	Classification State	ETDL	Energy Theft Detection Layer
*H*	Hour	$L_{d}$	Load Consumption Data
*M*	Month	$day_{d}$	*d*th day
SVC	Support Vector Classifier	ECPL	Energy Consumption Prediction Layer
∑	Summation	ET	Energy Theft
∀	for all	SM	Smart Meters
LIT	Linear Interpolation Technique	$D_{P}$	Preprocessed data
$C_{thres}$	Threshold Consumption	TEC	Threshold Energy Calculator
Pred	Predicted Load Values	$SM_{R}$	Meter readings
TL	Technical loss	NTL	Non-Technical Loss
NR	Normal Consumtion	γ	Loss Type
$\omega^{h_{i}}$	Load Consumption in Hour *i*	α	Threshold Consumption
θ	Comapartor Function	ξ	Threshold Limiter
Δ	Modulus Function	$SV_{l}$	SVC label
Φ	Loss Classifier	ψ	Energy
*C*	Set of Consumers	*A*	Set of Appliances
*S*	Set of Energy Sources	RF	Random Forest
LR	Linear Regression	-	-
LCL	Low-Carbon London	ETDM	Detection of Energy Theft and Defective Smart Meters
CNN	Convolutional Neural Network	CVLR	Categorical Variable-Enhanced Linear Regression
SEAI	Sustainable Energy Authority of Ireland	SMOTE	Synthetic Minority Over-Sampling Technique

**Table 2 sensors-22-04048-t002:** Comparative analysis between existing state-of-the-art work with our proposed work.

Author	Year	Objective	Pros	Cons
Wen et al. [19]	2018	Privacy preserving and energy theft detection in smart grids	Reduce the computation overhead and efficient energy theft detection	Proposed detection mechanism supports only linear systems
Sakhnini et al. [21]	2019	Detection of cyberattacks on smart grids using AI models	Improved detection rate and feature selection using genetic algorithm	Slow convergence because of a complex fitness function
Yan et al. [33]	2021	Electricity theft detection using AI models for smart meters	Consider six different types of attack in the dataset	Feature space is small and data imbalance problem
Lin et al. [36]	2021	Electricity theft detection using autoencoders and resampling techniques	Emphasis on detection strategies and solves class imbalance problem	Proposed model is not evaluated features specific to electrical data
Godahewa et al. [37]	2022	Optimize the energy consumption of an air conditioner using DL	Reduce the energy consumption to approximately 15–20%	Not consider the security aspect (energy theft)
Saoud et al. [38]	2022	Reducing the household energy consumption using wavelet transform and AI models	Improved prediction performance using wavelet transform	Complex reconstruction of the signals by wavelet transform
The proposed work	2022	Detection of energy theft using DL-based analytical scheme	Combinatorial approach of LSTM and support vector machine (SVM) enhances the detection rate	-

**Table 3 sensors-22-04048-t003:** *GrAb*: LSTM-based prediction model.

Energy	Consumption	Prediction Model	Structure
Attribute	Layer 1	Layer 2	Layer 3
Type	Input–LSTM	Hidden–LSTM	Output–Dense
No. of Nodes	50	50	1
Dropout	5%	5%	-
Model Type	Sequential		

**Table 4 sensors-22-04048-t004:** Prediction accuracy for individual devices in *GrAb*.

Device Name	MAPE Value (%)	RMSE Value
AC	22.82	0.0111
Lights	3.98	0.0148
Misc	0.28	0.0053
Basic Facilities	2.61	0.0256
Average	7.42	0.0137

**Table 5 sensors-22-04048-t005:** SVC model based Loss Classification Result.

Device	Time	Allowed Fluctuation (kWh)	Observed Fluctuation (kWh)	SVC Label	TL/NTL Count	Loss Type (T/NTL)
AC	3:00 a.m.	0.017	0.026	—1	2/2	TL
	4:00 a.m.	0.017	0.045	—1		
	8:00 p.m.	0.016	0.034	1		
	9:00 p.m.	0.019	0.031	1		
Lights	11:00 a.m.	0.001	0.064	1	02/01	TL
	12:00 p.m.	0.002	0.076	1		
	9:00 p.m.	0.012	0.158	—1		
Basic Facilities	7:00 a.m.	0.083	0.259	—1	0/4	NTL
	8:00 a.m.	0.019	0.16	—1		
	8:00 p.m.	0.008	0.166	—1		
	9:00 p.m.	0.013	0.198	—1		
Miscellaneous Facilities	6:00 a.m.	0.008	0.026	—1	0/2	NTL
	8:00 p.m.	0.011	0.070	—1		

## Data Availability

Not applicable.

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
