# Peer review of "GrAb*: A Deep Learning-Based Data-Driven Analytics Scheme for Energy Theft Detection"

_sensors, 2022, doi:10.3390/s22114048_

Round 1
Reviewer 1 Report
Dear Collegues
the paper is extremely clear and deals with a well known problem of the electrical energy thiefts which afflict all systems of production, trasportation and dispatch of electrical energy in all over the world. For this reason, the topic is not new and the ways to deal it are many as correctly reported in the introducion and in bibliography which, however, could have been wider.
As said the paper is clear in the mission and in its intrinsic set up that is well done making it never boring. The mathematical section is compact and clear, the experimental section is well conceived with results that justify the theoretical approach and which assume more strenght because compared with what already exist in literature, obtaining, obviously, better results.
In my opinion, the paper can be accepted in this form, but, in order to refine it, I only suggest two papers that the Authors can take into account for expanding the bibliography so giving a minor revision.
The first paper is:
1) Caciotta, M., Giarnetti, S., Leccese, F. Hybrid neural network system for electric load forecasting of telecomunication station (2009) 19th IMEKO World Congress 2009, 1, pp. 586-590. Cited 23 times. https://www.scopus.com/inward/record.uri?eid=2-s2.0-84871593385&partnerID=40&md5=3d1ab4231d2226852b06b88e6e628351
which, although, not perfectly centered on the problem of the electrical energy theft, is a good example of use of Neural Network used to forecast an electrical energy load. The use of NN is very complex and particular.
2) The second paper is very recent and discuss from a general point of view the problem of electrical energy theft showing the more recent perspectives.
Lin, Z., Cui, X., Jin, W., Liu, S., Feng, X., Feng, H. 22835623100;57215538133;57202822270;57201447108;35169220500;57223923287; Key Technologies of Electricity Theft Detection at Consumer Side [用户侧窃电检测关键技术] (2022) Dianli Xitong Zidonghua/Automation of Electric Power Systems, 46 (5), pp. 188-199. https://www.scopus.com/inward/record.uri?eid=2-s2.0-85126101866&doi=10.7500%2fAEPS20201010002&partnerID=40&md5=bb8c711a231c4eece2bb82b018d506d0 DOI: 10.7500/AEPS20201010002
Reviewer 2 Report
This article proposes a DL-based LSTM model for energy consumption prediction. The authors claim that their model can detect energy theft or technical losses. The idea of the paper is good, but the paper needs significant revision and improvement in both English writing and the presentation of materials, methods, and results. Here are some comments:
- More basic concepts have been focused in this paper. So needs to be modified in overall paper.
- My main concern is regarding that the comparison made by the authors has not on the basis of latest research findings. For the proposed compression method, various authors have developed latest energy consumption prediction in recent one or two year.
- Add the literature from recent studies in the field of energy consumption prediction and make the comparison with your techniques.
- Provide more details about the Table 3: Prediction accuracy for individual devices.
- Please explain the figure 5. RMSE values for devices using GrA with more details.
- “Here, the load consumption analysis for energy theft of Basic facility is shown in Figure10(b) and 10(a). In this case the number of hourly lower deviations exceeds that of higher deviations, hence detected as energy theft” could you please explain. It’s not very clear for me.
- Please provide more explanation of Figure 11 (I think it’s the most important contribution of the paper).
- Please add on brief note on future scope form the point of ‘applications-oriented research’ of the proposed work.
- Carefully noticed the sentence making and grammatical mistakes in the manuscript.
Format:
- Rewrite the introduction section.
- Replace the figure 5 (two-dimension bars with grid).
- At least 5 keywords should be associated to the field added in alphabetical order.
Reviewer 3 Report
Detecting energy theft in smart grids is certainly a very important task. The approach proposed by the authors is based on the comparison of the monitored energy consumption with an estimated value derived from historical data. Despite the approach being interesting, it is not possible to properly evaluate it due to a large number of open questions, and the quality of the presentation.
General remarks:
- figures and tables must be always reported after being mentioned in the text. Furthermore, a better placement should be considered, to avoid the usage of one entire page, for small figures;
- letters and symbols should not be mixed in the text. For example, the superscript “i” is used firstly to indicate the i-th hour of the day, then for the i-th appliance. Then, the generic appliance is also indicated as A_k. Sometimes the indices are also erroneously exchanged. For example, in Section 3.2 the appliances are numbered from 1 to n, while in section 4.1 they are numbered from 1 to m.
Section 3.1 :
- it is not clear how the Ld values have been normalized (with respect to what value?).
- more detailed considerations on the impact of the simple interpolation on the proposed technique should have been included. For example, in case of a long period of missing data, or even peaks, the linear interpolation cannot provide an accurate estimation.
- how is Ld composed? What is assumed to be the reporting rate of the considered monitoring devices? It looks like 1 value every hour. Which “energy consumption” quantities are considered?
- unique labelling should be used for the stages. At the beginning of the section, the stages are indicated as (i) and (ii), then with numbers (e.g. “stage 1”). Furthermore, the label should be also reported in Figure 3, for better understanding.
- in figure 3 it would be better to clearly indicate which parts belong to GrAb (with a background coloured area)
- Tale 1 must be reported at the beginning of the paper, not in the middle when most of the symbols have been already used.
Section 4:
- not clear why in this section the GrAb is divided into four layers. These layers seem to be a further division of the stages described in section 3.1. The authors should better clarify what are the differences and add an indication of these layers also in Figure 3.
Results:
- is not clear how the MAPE and the RMSE have been evaluated to define the accuracy of the prediction. Additional information should be reported on how many points have been considered for each appliance and so on;
- the authors should clearly specify how the false data have been “injected”. Was a Monte Carlo approach considered? What are the characteristics of the distributions used to extract the new data, and so on;
- the figures are not well explained and it is difficult to understand exactly what they represent and how “read” them to verify what is reported in the text.
Round 2
Reviewer 2 Report
The authors have addressed my concerns, and I recommend publication in the journal.
Reviewer 3 Report
This reviewer would like to thank the authors for the effort in improving the paper. Nevertheless, two previous comments have not been addressed, and further actions are still required.
Previous comment 2, the indices in the equations have not been fixed. For example, in eq. (1) the index of the i-th hour is still wrong. Instead of “i” authors reported “y”. Further errors are reported in the other equations and in the related text.
Previous comment 5, based on the information provided by the authors, it is not possible to exactly determine which database has been selected (among all of the available ones in [44]). Thus, authors should clearly report which database has been used, together with the metrological characteristics. In particular, important information such as reporting rate, or considered sampling frequency, must be reported with reference to the current paper. Again, from the text, one could guess that 1 measurement per hour has been considered, but it is not clear if this is the same reporting rate of the original database or if the authors have considered a down-sampling. In case of the latter, considering the full set of data would have allowed the authors to not consider the injection of synthetic data for the training, and would have given a higher validity to the training.
In Figures 8 to 11, the subfigures (a) and (b) could be merged into a unique figure, since the baseline is the same. This would facilitate the comparison between the normal load profile and the anomalous load profile.
